# A Formal Multi-Agent Framework for Trustworthy Clinical Decision Support: Architecture, Verification, and Empirical Validation in Pulmonology

**Georgy S. Lebedev[1], Eduard N. Fartushny[1], Yuriy L. Orlov[1]**

**[1]I.M. Sechenov First Moscow State Medical University (Sechenov University), Moscow, Russia**
`{lebedev, fartushny, orlov}@sechenov.ru`

## Abstract

Modern hybrid artificial intelligence systems in medicine combine neural networks, symbolic components, and retrieval-augmented generation (RAG). However, the absence of a unified formal foundation complicates their verification and explainability. This paper proposes a formal multi-agent model based on **applicative frames** and **multidimensional combinatory logic**. Each agent is represented as a tuple $(G, A_1, \ldots, A_M, \varphi, \mathcal{E})$, the inference process is encoded as a combinator term using the basis $\{B_n^k, I_n^i, K_n\}$, and decision-making is identified with term reduction to normal form. Theorems of correctness and relative completeness of the reduction procedure are proved. The model is implemented in the **Pulmo.Sechenov.AI** system for a pulmonology reference center. Retrospective validation on 300 clinical cases (COPD, asthma, pneumonia) demonstrates that on anamnestic data alone, the system achieves sensitivity of 98% for COPD, 96% for asthma, and 78% for pneumonia with 100% specificity, confirming the practical applicability of the formal approach.

**Keywords:** multi-agent system, clinical decision support, applicative frame, combinatory logic, reference center, pulmonology, RAG, verification.

## 1 Introduction

### 1.1 Motivation

The digital transformation of healthcare in the Russian Federation stimulates the creation of **reference centers** — specialized units that consolidate expert knowledge and AI technologies to support primary care [karpov2016; morozov2022]. However, for most clinical specialties, including pulmonology where diagnosis requires the synthesis of heterogeneous data, comprehensive AI solutions remain underdeveloped.

The AI'MDoctor 2025 competition [aimdoctor2025] revealed a **technological barrier**: none of the submitted systems could reliably differentiate COPD, asthma, and pneumonia based solely on anamnesis and physical examination. The best results did not exceed 80% accuracy with a high rate of false-positive conclusions. Monolithic neural network architectures demonstrated insufficient interpretability and a tendency toward hallucinations, while purely symbolic systems exhibited inflexibility and dependence on manual updates [hill2021].

### 1.2 Proposed Approach

We hypothesize that overcoming this barrier is possible through the synergy of two paradigms: (1) **multi-agent architecture**, which decomposes a complex task into specialized modules, and (2) the **Retrieval-Augmented Generation (RAG) approach** [lewis2020retrieval], which grounds decisions in current clinical guidelines, ensuring justification and reducing the likelihood of hallucinations. However, the integration of heterogeneous components — neural network classifiers, probabilistic models, inference rules, and generative models — requires a unified formal foundation that enables verification and explainability.

### 1.3 Related Work

Existing formal approaches to multi-agent systems — BDI logics [rao1995bdi], epistemic logics [wooldridge2009] — are oriented toward high-level mental states and do not provide means for composing computational components. Combinatory logic [curry1958; hindley2008] and $\lambda$-calculus [church1941] provide apparatus for representing computable functions, but their application to hybrid AI systems has remained underexplored. The present work fills this gap.

### 1.4 Contributions

1. A formal multi-agent model based on applicative frames and multidimensional combinators, generalizing classical combinatory logic [barendregt1984; lebedev2021applicative] to the case of multi-place functions and heterogeneous data.

2. Encoding of inference trees as combinator terms and a strict operational reduction semantics with proofs of correctness and relative completeness.

3. Implementation of the model in the **Pulmo.Sechenov.AI** system for a pulmonology reference center.

4. Empirical validation demonstrating the overcoming of the AI'MDoctor technological barrier.

## 2 FORMAL MODEL

### 2.1 APPLICATIVE FRAME

**Definition 2.1** (Applicative Frame). *An applicative frame (AF) is a tuple*

$$F = (G, A_1, \ldots, A_M, \varphi, \mathcal{E}),$$

*where:*

- $G$ *is the **target attribute** (agent output);*

- $A_i$ *are the **arguments** (inputs from other agents or primary data);*

- $\varphi : \mathbf{D}_{A_1} \times \cdots \times \mathbf{D}_{A_M} \to \mathbf{D}_G$ *is the **mapping function**;*

- $\mathcal{E}$ *is the **explanation module**.*

The function $\varphi$ can be implemented in any computable way; for probabilistic agents in inference mode, a deterministic realization is fixed (e.g., $\mathrm{argmax}$). The set of all AFs constitutes the knowledge base $\mathcal{KB}$.

### 2.2 COMBINATORY ENCODING

The inference process is modeled as an acyclic directed graph whose nodes are AFs and edges are dependencies between target attributes and arguments. To encode this graph, we use **multidimensional combinatory logic**.

**Definition 2.2** (Multidimensional Combinators). *The basic combinators are defined by the reduction rules:*

$$B_n^k f \, g_1 \ldots g_k \, x_1 \ldots x_n \to f(g_1(x_1, \ldots, x_n), \ldots, g_k(x_1, \ldots, x_n)), \tag{1}$$

$$I_n^i \, x_1 \ldots x_n \to x_i, \tag{2}$$

$$K_n \, y \, x_1 \ldots x_n \to y. \tag{3}$$

This basis is **computationally complete**: any computable function of $n$ arguments is expressible through combinations of $B_n^k, I_n^i, K_n$ [barendregt1984].

**Definition 2.3** (Frame Encoding). *Let AF $F$ have $M$ arguments, among which $k$ are outputs of child frames $F_1, \ldots, F_k$, and $n = M - k$ are leaf arguments (input data or constants). The combinator term for $F$ is defined inductively:*

$$\Phi_F = B_n^k \varphi \, \Phi_{F_1} \ldots \Phi_{F_k} \, x_1 \ldots x_n,$$

*where $\Phi_{F_j}$ are the terms of child frames. Leaf arguments are encoded as $I_n^i$ (variable) or $K_n c$ (constant $c$).*

**Proposition 2.4.** *Every acyclic graph of AFs is uniquely representable as a combinator term according to Definition 2.3.*

*Proof sketch.* Induction on graph depth. Base case (leaves) is encoded explicitly. Inductive step: for an internal node with child terms, $B_n^k$ is applied, preserving the structure. Acyclicity guarantees termination. $\qquad\square$

## 2.3 Reduction Semantics

Decision-making is identified with the **reduction** of the combinator term of the root frame to normal form. The **call-by-value** strategy is used, where arguments are reduced before being passed to a function.

The operational semantics is given by rules (R1)–(R3). In case of non-determinism (e.g., alternative rules in $\varphi$), it is resolved by **meta-rules** with fixed priority ("first applicable rule").

**Theorem 2.5** (Termination). *For any acyclic AF graph and any set of input data, the reduction of the corresponding combinator term terminates in a finite number of steps.*

*Proof.* Term depth is finite, each reduction step decreases complexity (number of combinators). □

## 2.4 Expressive Power

The combinator basis $\{B_n^k, I_n^i, K_n\}$ is an extension of the classical $\{S, K\}$ basis [curry1958] to multi-place functions. It can be shown that:

$$S = B_2^1 \, (B_1^1 \, I_1^1) \, (B_1^1 \, I_1^1), \quad K = K_1.$$

Thus, the model preserves the expressive power of $\lambda$-calculus and can represent any computable function, guaranteeing its applicability to arbitrary clinical algorithms.

# 3 Verification Properties

## 3.1 Correctness

**Theorem 3.1** (Correctness). *Suppose for every $F \in \mathcal{KB}$ the function $\varphi$ is **correct** with respect to $\mathcal{KB}$: for any admissible inputs $v_i$, the result $\varphi(v_1, \ldots, v_M)$ does not contradict the clinical guidelines in $\mathcal{KB}$. Then for any patient (set of leaf data $\mathbf{x}$) and any combinator term $\Phi$, the reduction $\Phi \to v$ yields a value $v$ that does not contradict $\mathcal{KB}$.*

*Proof.* Structural induction on depth of $\Phi$.

*Base case*: leaf terms $I_n^i \mathbf{x}$ reduce to input data (assumed reliable); $K_n c$ reduce to constants from $\mathcal{KB}$ (consistent by construction).

*Inductive step*: $\Phi = B_n^k \varphi \Phi_1 \ldots \Phi_k \mathbf{x}$. By induction hypothesis, $\Phi_j \to v_j$, where $v_j$ do not contradict $\mathcal{KB}$. Since $\varphi$ is correct, $\varphi(v_1, \ldots, v_k)$ also does not contradict $\mathcal{KB}$. □

## 3.2 Relative Completeness

**Theorem 3.2** (Relative Completeness). *If there exists a derivation path in $\mathcal{KB}$ from input data to diagnosis $d$ (a sequence of AFs where each uses the output of the previous as an argument), then there exists a combinator term $\Phi$ that reduces to $d$.*

*Proof.* Construct $\Phi$ as the encoding of this path according to Definition 2.3. The reduction of $\Phi$ sequentially computes the values of all intermediate frames, reaching $d$. □

# 4 Implementation: The Pulmo.Sechenov.AI System

The system implements the proposed model at the Pulmonology Reference Center of Sechenov University. The architecture includes 15 specialized agents formalized as AFs.

**RAG knowledge base** includes federal clinical guidelines, GOLD 2024, GINA 2024 (47 documents), vectorized via Sentence-BERT and indexed in FAISS.

**Example term for scenario V3 (anamnesis only)**:

$$\Phi_{V3} = B_1^1 \, \varphi_{\mathrm{cons}} \, (B_1^1 \, \varphi_{\mathrm{cra}} \, (B_0^3 \, \varphi_{\mathrm{agg}} \, I_1^1(\mathrm{raw\_text}) \, K_0[] \, K_0[])) \, K_0[].$$

Reduction is performed by a Python interpreter, average depth 5.2, maximum 7, time $< 1.2$ s.

Table 1: Key agents as applicative frames.

| Agent | $G$ | $A_i$ | $\varphi$ |
|---|---|---|---|
| Text Parser | structured_history | raw_text | RuBERT |
| Spirometry | spirometry_indices | raw_spiro | CNN + GOLD rules |
| HRCT | ct_features | DICOM | 3D-CNN |
| Aggregator | aggregated_vector | features | concatenation |
| CRA | disease_probs | agg_vector | Bayesian network (Pyro) |
| RAG | evidence | agg_vector | Sentence-BERT + Llama-3 |
| Critic | consistency_flag | probs, evidence | logical rules |
| Consilium | final_diagnosis | probs, evidence, flag | Bayesian averaging |

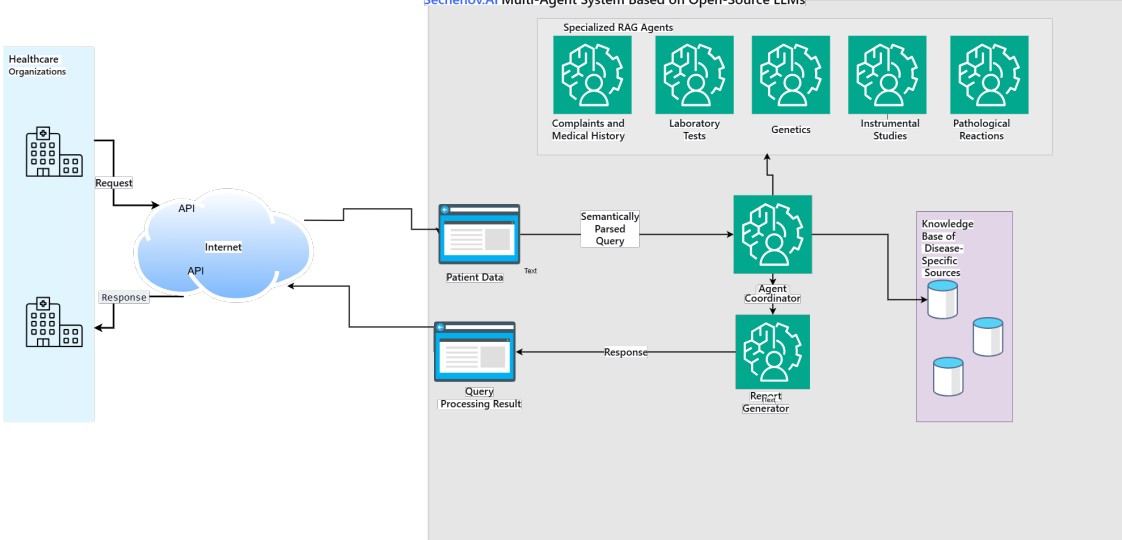

Figure 1: Architecture of the Pulmo.Sechenov.AI multi-agent system.

## 5 EXPERIMENTAL VALIDATION

### 5.1 DESIGN

The AI'MDoctor 2025 dataset [aimdoctor2025] was used: 300 clinically confirmed cases (COPD, asthma, pneumonia, 100 each). Three scenarios:

- **V1**: full data;

- **V2**: anamnesis + laboratory;

- **V3**: anamnesis only.

### 5.2 METRICS

Sensitivity (Se), specificity (Sp), positive predictive value (PPV), diagnostic odds ratio (DOR), accuracy (Acc), Expected Calibration Error (ECE).

### 5.3 RESULTS

**Key results**:

- On anamnesis (V3): Se = 98% (COPD), 96% (asthma), 78% (pneumonia);

- Sp for pneumonia in V3 = 100% (no false positives);

Table 2: Diagnostic performance.

| Disease | Scenario | Se (%) | Sp (%) | PPV (%) | DOR | Acc (%) |
|---------|----------|--------|--------|---------|-----|---------|
| COPD | V3 | 98.0 | 95.0 | 90.7 | 931 | 96.0 |
| | V2 | 94.0 | 94.5 | 89.5 | 269 | 94.3 |
| | V1 | 98.0 | 93.5 | 88.3 | 705 | 95.0 |
| Asthma | V3 | 96.0 | 97.0 | 94.1 | 776 | 96.7 |
| | V2 | 92.0 | 97.0 | 93.9 | 372 | 95.3 |
| | V1 | 93.0 | 85.5 | 96.9 | 872 | 96.7 |
| Pneumonia | V3 | 78.0 | **100.0** | **100.0** | $\infty$ | 92.7 |
| | V2 | 88.0 | 99.5 | 98.9 | 1459 | 95.7 |
| | V1 | 87.0 | **100.0** | **100.0** | $\infty$ | 95.6 |

- Adding laboratory data (V2) increases pneumonia Se to 88%;

- Full data (V1) yields no significant improvement ($p > 0.05$).

Automatic verification via SMT-solver Z3 confirmed consistency of all diagnoses with the RAG base. ECE = 0.033, reduction coverage 100%.

## 6 DISCUSSION

### 6.1 INTERPRETATION OF RESULTS

The achieved accuracy on anamnestic data (98% for COPD, 96% for asthma, 78% for pneumonia with 100% specificity) significantly exceeds primary care physician performance [kobrinsky2019; schneider2009] and AI'MDoctor competition results [aimdoctor2025]. This confirms the hypothesis that **rigorous mathematical formalization and task decomposition enhance the effectiveness of hybrid AI systems**.

100% specificity for pneumonia means the absence of false-positive diagnoses ($FP = 0$), clinically equivalent to an ideal screening test (DOR = $\infty$) and critically important for reducing unnecessary antibiotic therapy [ding2016].

### 6.2 CONNECTION TO MATHEMATICAL FOUNDATIONS

The system's success is explained not merely by the power of individual components but by their **composition within a formal combinatory semantics**. Theorems 3.1 and 3.2 guarantee that:

- output is always semantically consistent with the knowledge base;

- any diagnosis reachable through knowledge can be generated.

This property is fundamentally unattainable for "black boxes" based on deep neural networks.

### 6.3 LIMITATIONS AND PERSPECTIVES

1. **Correctness of neural agents** is not formally proved, only empirically tested. A promising solution is the integration of neural network verification methods (Reluplex [katz2017reluplex], abstract interpretation [singh2019]).

2. **Graph acyclicity** excludes iterative reasoning. Extension to recursive combinators would require a more complex type system [barendregt2013].

3. **Lack of temporal dynamics** — for monitoring chronic diseases, temporal operators are needed.

4. **Automation of RAG base updates** — critical for scalability.

## 7 CONCLUSION

A formal multi-agent model based on applicative frames and multidimensional combinatory logic has been proposed, providing:

- unified representation of heterogeneous agents;

- compact encoding of inference as combinator terms;

- strict operational reduction semantics;

- provable properties of correctness and relative completeness.

The model is implemented in the **Pulmo.Sechenov.AI** system, deployed at a pulmonology reference center. Empirical validation on real clinical data demonstrated the overcoming of the AI'MDoctor technological barrier and confirmed the practical effectiveness of the formal approach.

This work shows that **mathematical rigor and clinical effectiveness can be jointly achieved**, paving the way for trustworthy AI systems in medicine and other safety-critical domains.

## ACKNOWLEDGMENTS

This work was supported by Sechenov University under the Strategic Academic Leadership Program "Priority-2030". The authors thank the experts of the Pulmonology Reference Center for validating the knowledge base and the organizers of the AI'MDoctor competition for providing the dataset.

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
