# OpenReview forum: "A Formal Multi-Agent Framework for Trustworthy Clinical Decision Support: Architecture, Verification, and Empirical Validation in Pulmonology"
_mathai.club/MathAI/2026/Conference — MathAI 2026 Conference Submission_

### Official Review · Reviewer_yBJp · 2026-03-12
**A FORMAL MULTI-AGENT FRAMEWORK FOR TRUSTWORTHY CLINICAL DECISION SUPPORT: ARCHITECTURE, VERIFICATION, AND EMPIRICAL VALIDATION IN PULMONOLOGY**

**Rating:** 2
**Confidence:** 5

**Review:**

This article is devoted to the development of a multi-agent model for a clinical decision support system in pulmonology. The authors propose using frames and combinatorial logic to formalize the inference process in a hybrid AI system combining neural network components, symbolic rules, and RAGs.
Strengths

1) The problem of interpretability and verifiability of hybrid AI systems in medicine is truly relevant. The paper identifies the limitations of monolithic neural network architectures (prone to hallucinations, insufficient explainability) and purely symbolic systems (inflexibility, dependence on manual updates). The authors attempt to describe a unified formal approach for creating heterogeneous components.

2) The diagnostic task is carefully decomposed into 15 specialized agents with clearly defined inputs and outputs. The presence of a critical agent and a "consultation" agent provides elements of self-checking for the system.

Weaknesses

1) A serious problem with the article is the gap between the presented formalism and its implementation. The apparatus of multivariate combinatorial logic is introduced, and theorems of correctness and relative completeness are proved, but the connection with the actual operation of the system remains unclear.

2) The theorems and proofs are unclear. Theorem 3.1 (correctness) is essentially a tautology: if each component is correct, then the composition is correct. This is true for any deterministic system and does not require the apparatus of combinatorial logic. Theorem 3.2 (relative completeness) merely states that if an inference path exists, it can be encoded, which is a consequence of the definition of encoding in Definition 2.3, not a substantive result. The proofs are very superficial and amount to a single sentence.

3) Experimental validation was conducted on 300 cases, 100 for each disease. This is insufficient for the stated conclusions. With 100 COPD cases, a sensitivity of 98% means the system made only two errors, and a difference of one error would change the result by a full percentage point. The authors do not provide confidence intervals.

4) There is no comparison with other approaches to multi-agent diagnostics, with ensemble machine learning methods, or even with individual system components (ablation study).

5) The RAG knowledge base contains 47 documents. This is a very modest volume, and the update procedure is not described.

6) The reference list contains references unrelated to the article's content. For example, [hill2021] is a paper on insurance, [ding2016] is an article on endobronchial valves, both of which are unrelated to multi-agent systems or COPD/asthma/pneumonia diagnostics. This raises questions about the article's preparation.

Conclusion

In its current form, the article cannot be recommended for publication. Furthermore, the authors failed to comply with the terms and conditions and did not remove their names from the title.

---

### Official Review · Reviewer_RRer · 2026-03-12
**Review of "A Formal Multi-Agent Framework for Trustworthy Clinical Decision Support: Architecture, Verification, and Empirical Validation in Pulmonology"**

**Rating:** 3
**Confidence:** 4

**Review:**

The strengths of the paper lie in the effective decomposition of the diagnostic task into 15 specialized agents and the use of modules to ensure logical consistency, though a gap remains between the complex formalism and its practical implementation.

Weaknesses include the loose integration of combinatory logic into the system and the trivial nature of the proven theorems. Furthermore, the empirical sample of 300 cases is insufficient to validate the reported high accuracy without confidence intervals or comparative analysis, necessitating major revisions before publication.